# Experimental Investigation on the Effect of Salt Solution on the Soil Freezing Characteristic Curve for Expansive Soils

Haiwen Yu [1,2], Fengfu Hao [3], Panpan Yi [1], Qin Zhang [3] and Tiantian Ma [1,*]

1 State Key Laboratory of Geomechanics and Geotechnical Engineering, Institute of Rock and Soil Mechanics, Chinese Academy of Sciences, Wuhan 430071, China; yuhaiwen20@mails.ucas.ac.cn (H.Y.); ppyi@whrsm.ac.cn (P.Y.)
2 University of Chinese Academy of Sciences, Beijing 100049, China
3 Guangxi Key Laboratory of Geomechanics and Geotechnical Engineering, College of Civil Engineering and Architecture, Guilin University of Technology, Guilin 541004, China; fengfuhao1220@163.com (F.H.); ydyxzhangqin@163.com (Q.Z.)
* Correspondence: ttma@whrsm.ac.cn; Tel.: +86-136-4721-1718

**Abstract:** With the development of the Belt and Road Initiative in China, high-speed railways are booming and inevitably pass through seasonal frost regions. In Nanyang basin, due to seasonal changes, railway subgrades will undergo frost heaving and thawing subsidence. The freezing characteristics of the soil are characterized by the freezing characteristic curve, and the important factors affecting the freezing characteristic curve are the content of expansive clay minerals in the soil and the salt solution. Therefore, three soil samples with different montmorillonite contents were saturated with salt solutions of different concentrations, and the freezing temperature of the soil samples was controlled by a cold bath. After the temperature equilibrium, the frozen stable soil samples were put into a nuclear magnetic resonance instrument to test the unfrozen water content, and the relationship between the freezing temperature and the unfrozen water content of expansive soil under different salt solution concentrations was obtained. Additionally, a unified model was used to simulate the test results. The results show that SFCC shifts to the left, that is, the sodium chloride salt solution reduces the freezing point of the soil sample so that it has more unfrozen water at the same temperature. At the same time, the soil's freezing characteristic curves are closely related to content of expansive clay minerals in the soil. The more expansive clay mineral content, the greater the corresponding unfrozen water content. These results provide some basic insights for improving the frost heave and thaw subsidence problems of railway subgrades in seasonal permafrost regions, which will have a positive impact on promoting the management and rational application of land resources and the promotion of sustainable development.

**Keywords:** expansive soil; illitic soil; sodium bentonite; salt solution; freezing characteristic curve





## 1. Introduction

Permafrost and seasonal frost cover 21.5% and 53.5% of China's land area, respectively [1,2]. With the development of China's "Belt and Road" initiative, more and more high-speed railways have been opened in permafrost areas in recent years [3,4]. In seasonal frost regions, the subgrade soil will undergo frost heaving at low temperatures and thaw subsidence at temperatures above 0 °C. This seriously affects the safety of railway train operations [5,6]. The freezing and thawing of soil, accompanied by the phase change between ice and water, involves the release or absorption of latent heat by the soil. This process impacts the energy balance at the Earth's surface [7–9]. In recent years, remote sensing technology has been applied to the identification of permafrost regions; satellite images of the surface temperature can be used to identify isothermal periods of the ground temperature during seasonal freeze and thaw and provide the necessary data for numerical modeling of the thermal state of the ground [10]. In the final analysis, a thorough

understanding of the characteristics of frozen soil can positively impact our ability to better manage and utilize land resources, accurately assess the risks of frost heave and thaw settlement in seasonal frozen areas, protect the environment, preserve biodiversity, and promote sustainable development.

Unlike free water, not all pore water in the soil will freeze when the temperature is below zero, resulting in the coexistence of ice and liquid water in soil pores at temperatures below 0 °C. This is mainly caused by three factors: (a) capillary and adsorption effects (Spaans and Baker, 1996 [11]; Zhou et al., 2018 [12]), (b) the solute effect, where the presence of salt in the soil lowers the freezing temperature [13], and (c) granularity: namely, with the increase in fine particle content in the soil, the water migration ability in the soil during the freezing process is enhanced, and the frost heaving property of the soil is enhanced [14–19]. The relationship between unfrozen water content and temperature is characterized by a freezing characteristic curve (SFCC). The unfrozen water in frozen soil is the channel for water migration, and its mechanical properties are also closely related to the content of unfrozen water [20–22]. Some scholars have pointed out that the migration of fine particles to the warm parts of the soil is caused by electrophoresis caused by frost (Bertouille, 1972 [23]). Therefore, in the study of frozen soil, the freezing characteristic curve is an important constitutive relationship. SFCC is similar to the soil–water characteristic curve (SWCC) in unsaturated soil, which characterizes the relationship between water content and matrix potential in unsaturated soil (Flerchinger et al., 2006 [24]; Koopmans and Miller, 1966 [25]; Spaans and Baker, 1996 [11]).

The main factors affecting soil freezing characteristics are soil properties (particle composition, liquid and plastic limits, mineral composition) and the salt solution [26]. Wang et al. [27] proposed a model that considers the geometric shape and size of pores to predict the freezing characteristic curve of soil, and found that the residual water content is positively correlated with the clay content. Tsytovich [26] established the experimental and theoretical relationships between unfrozen water content and the plastic index, plastic limit and negative temperature. Chen et al. [28] found through testing frost samples from the Qinghai-Tibet Plateau that the content of unfrozen bound water gradually increases with the increase in clay or clay mineral content, and at a certain temperature, the content of unfrozen capillary water increases with the increase in silt content. A large number of research results show that salinity is the main factor controlling SFCC under given soil conditions. As the concentration of the salt solution increases, the freezing point will decrease [29–31], which can be explained by the Van't Hoff equation [13,32]. However, there are few studies on the freezing characteristics of soil under the interaction of expansive clay minerals and the salt solution.

Nanyang Basin is a seasonal frost belt, which is one of the main distribution areas of expansive soil in China, and salinization occurs in some areas. As Nanyang Basin is a typical distribution area of expansive soil and the starting point of the South-to-North Water diversion project, with many construction projects planned in the future, it is of great significance to study the effect of the salt solution on the freezing characteristic curve of Nanyang expansive soil. Therefore, this study uses medium expansive soil from the Nanyang area as the research object, and additionally selects sodium bentonite and commercial illitic soil for comparison, uses a cold bath to control the temperature of the soil sample, and uses nuclear magnetic resonance technology to obtain the unfrozen water content at the corresponding temperature. The influence law of content of expansive clay and NaCl solution concentration on the freezing characteristic curve of expansive soils is obtained, and the theoretical model proposed by Zhou et al. [12] is used to compare with the experimental results to verify its applicability in calculating the freezing characteristic curve of saline soils.

## 2. Materials and Methods

### 2.1. Materials

The experimental materials selected were medium expansive soil (NY) from the Nanyang area, sodium bentonite, and commercial illitic soil. Table 1 shows the physical and mechanical properties of the three soils. The reason for choosing expansive soil as the research object was that expansive soil contains expansive minerals (montmorillonite) and has strong water retention. At the same water content, the matrix potential is lower, so the corresponding freezing temperature is also lower, and the current understanding of the effect of the content of montmorillonite in the pore salt solution and soil on the freezing characteristics of expansive soil is still insufficient.

**Table 1.** Physical parameters of bentonite, NY, and illitic soil.

| Soil Type | Liquid Limit/% | Plastic Limit/% | Plasticity Index | Specific Gravity |
|---|---|---|---|---|
| Bentonite | 176.61 | 41.46 | 135.16 | 2.72 |
| NY | 57.4 | 26.5 | 30.9 | 2.73 |
| Illitic soil | 40.62 | 23.61 | 17.01 | 2.72 |

The particle size curves of the three soil samples are shown in Figure 1. In order to reduce the influence of particle size on the freezing characteristic curve, all three soil samples were crushed and sieved through a 0.1 mm sieve. Table 2 shows the mineral composition tables of three soil samples, which were obtained from mineral X-ray diffraction analysis tests. The montmorillonite content of bentonite, NY, and illitic soil was 68.3%, 33.7%, and 0%, respectively. Therefore, these three types of soil samples could be used to study the effect of montmorillonite content on the freezing characteristics of soil.

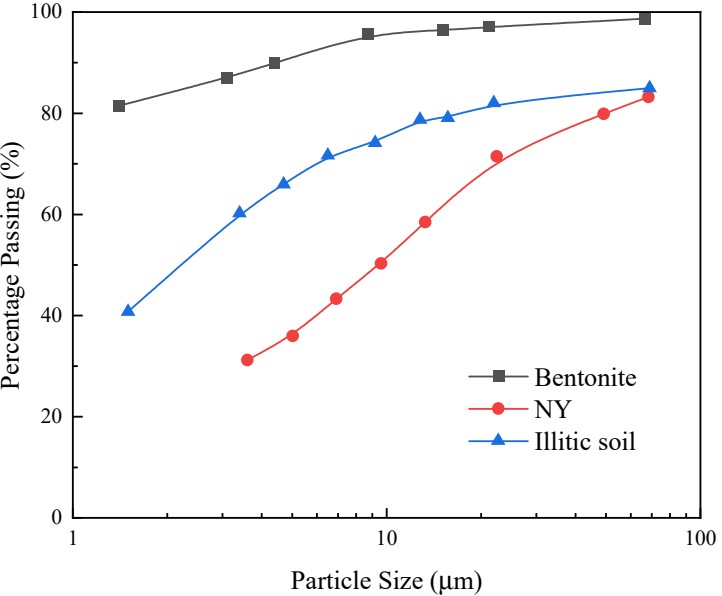

**Figure 1.** The particle size curves of bentonite, NY, and illitic soil.

**Table 2.** Mineral composition of bentonite, NY, and illitic soil.

| Soil Type | Clay Mineral Composition/% | | | | | |
|---|---|---|---|---|---|---|
| | Montmorillonite | Quartz | Illite | Albite | Microcline | Anorthite |
| Bentonite | 68.3 | 0.8 | - | - | - | 30.9 |
| NY | 33.7 | 29.7 | - | 24.9 | 11.7 | - |
| Illitic soil | - | 18.2 | 49.0 | 7.9 | 24.9 | - |

First, the moisture content of the air-dried soil was set to 20%, which was the best moisture content of NY measured by the compaction test. Then, the prepared sample was placed in a sealed plastic bag for more than 48 h to ensure that the water content of the sample was evenly distributed. The sample preparation method used static pressure to prepare samples. The dry density of NY and illitic soil was set to 1.5 g/cm$^3$. In the experiment, it was found that if the dry density of bentonite was set to 1.5 g/cm$^3$, due to its high expansibility [33], part of the soil sample was extruded from the saturator during the saturation process, and due to its low permeability, it was difficult for the salt solution to penetrate into the bentonite and fill its pores with brine. Therefore, we could only set the dry density of bentonite to 1.2 g/cm$^3$. If the dry density of illitic soil was also adjusted to 1.2 g/cm$^3$, the strength of the illitic soil sample would be too low after saturation, and it would assume a flowing state, which would make it difficult to maintain its stable state. Therefore, we finally set the dry density of bentonite to 1.5 g/cm$^3$ and the dry density of the other two soils to 1.2 g/cm$^3$. The size of the soil sample was 4.5 cm in diameter and 2 cm in height. In order to avoid interference from the ferromagnetic ring with the nuclear magnetic resonance signal, the ring here was made of polytetrafluoroethylene material that had no nuclear magnetic resonance signal and thus would not interfere with the magnetic field. Different concentrations of NaCl solution were selected for the salt solution, of 0, 0.1, 0.2, 0.5, 1, 2, and 5 mol/L; all were analytically pure, where 0 mol/L was deionized water. The pressed soil sample was placed in a vacuum cylinder under vacuum for 4 h, then immersed in a salt solution of corresponding concentration to submerge the soil sample, after which vacuum was continued for 4 h to make sure the sample was fully saturated.

## 2.2. Method

Nuclear magnetic resonance (NMR) measures the signal of hydrogen protons in pore water to obtain the water content in soil samples [34]. Hydrogen protons in water have non-zero spin and inherent magnetic moment. When placed in an external magnetic field, they spin around the external magnetic field orientation and precess. When irradiated with a 90° radio frequency pulse, the protons are excited and rotate, perpendicularly to the initial direction, causing magnetization. After turning off the radio frequency pulse, the spin relaxes to the equilibrium direction, and the apparent magnetization strength caused by the radio frequency pulse decays over time. The measured signal is called free induction decay (FID), where the maximum value corresponds to the number of hydrogen nuclei [35]. For a fixed mass of soil sample, this can obtain the water content of the soil sample. When pore water freezes into ice, no signal can be measured by NMR due to rapid relaxation. Therefore, unfrozen water content in the soil samples can be obtained based on NMR technology. This method has the advantages of being fast and non-destructive, and is often used in earth science to test pore size distribution in porous media and pore water interaction. Figure 2 shows the cold bath and NMR equipment used in the experiment.

To avoid supercooling effects, the temperature of the soil sample was gradually increased from the lowest temperature, and the unfrozen water content was measured during the melting process. The relationship between temperature and unfrozen water content, that is, the freezing characteristic curve, could be obtained. After the soil sample was saturated, it was removed from the water bath, excess water around the ring was wiped off, and the soil sample was put on a balance to weigh its saturated mass. Then, the soil sample was put into a cold bath with a temperature of −10 °C. The cold bath temperature slowly rose from −10 °C to 25 °C. When the soil sample reached equilibrium at each temperature level, a thermal parameter instrument was used to measure the temperature of the soil sample. At each temperature level, when the soil sample reached the freezing equilibrium state (about 4 h), it was removed and placed into the nuclear magnetic resonance test system for unfrozen water content measurement. The test results could be analyzed through inversion software that was included with the NMR equipment.

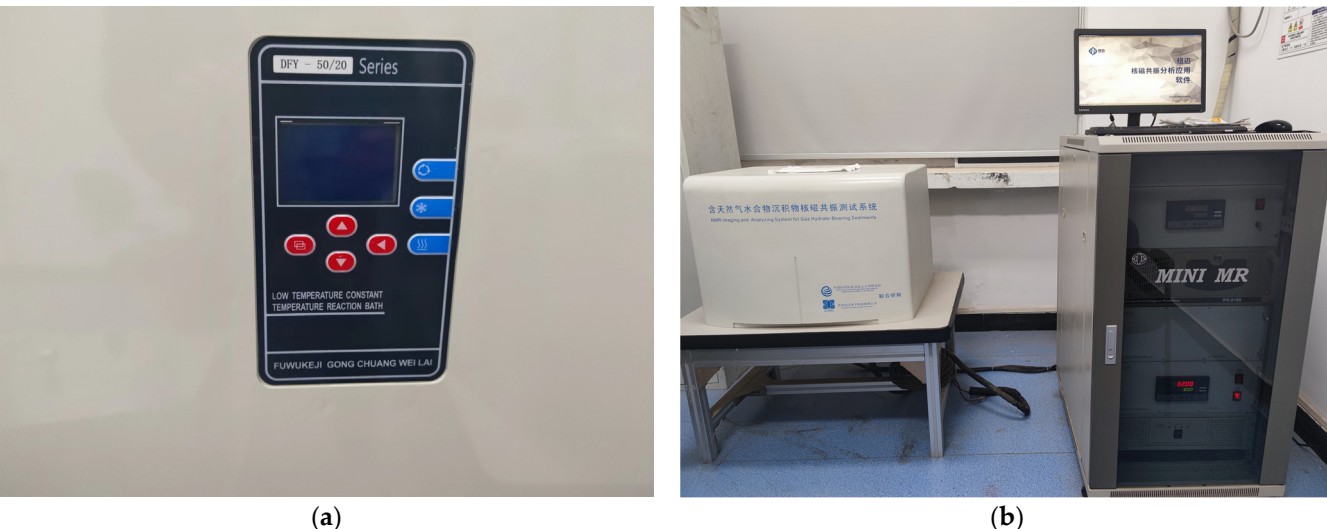

**Figure 2.** Experiment instruments. (**a**) Cold bath device; (**b**) Nuclear magnetic resonance instrument.

Figure 3 shows how to calculate the unfrozen water content. According to Curie's Law [36,37], the value of the magnetic resonance signal will increase as the temperature decreases, although the water content is constant. When the temperature is below zero, pore water begins to freeze and the magnetic resonance signal decreases dramatically. As shown in Figure 3, the paramagnetic response regression line is obtained by fitting the relationship between nuclear magnetic signal and temperature. Assuming that Curie's Law applies in the subzero temperature zone, the initial water content of the sample at $-5\ °C$ has a magnetic resonance signal value of $Y_1$. Combining the linear relationship between the magnetic resonance signal and water content, the unfrozen water content at $-5\ °C$ $w = (Y_2/Y_1) \times w_0$, where $w_0$ is the initial water content and $Y_2$ is the measured value of magnetic resonance signal [29].

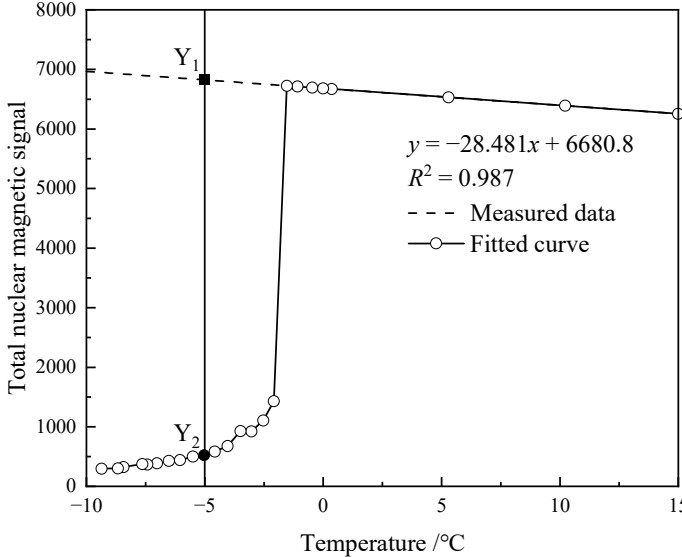

**Figure 3.** Calculation principle for unfrozen water content [29].

## 3. Results and Discussion

### 3.1. The $T_2$ Distribution Curves for Soils at Different Temperatures

Figure 4 shows the $T_2$ distribution curves of the three experimental soil samples under different temperatures, with the initial three samples saturated by deionized water. The black line is the saturated state at room temperature, the red line is the state when the

soil sample starts to freeze, and the blue line is the state when only residual unfrozen water remains in the soil sample pores. As the temperature decreases, the peak area of the $T_2$ distribution curve gradually decreases, indicating that after freezing, the liquid water content in the pores of the soil sample gradually decreases, and the measured nuclear magnetic signal quantity gradually decreases. In addition, as the pore water in the soil sample freezes, the relaxation time corresponding to the peak value of the $T_2$ distribution curve gradually decreases (the peak shifts to the left), indicating that water freezing in the soil first occurs in large pores. Scholars have pointed out that with the increase in the number of freeze–thaw cycles, soil particles will produce aggregates, that is, under the action of freeze–thaw cycles, compaction between soil particles will occur [38]. The corresponding results in the $T_2$ distribution curve are as follows: with the increase in the number of freeze–thaw cycles, the $T_2$ curve gradually changes from a unimodal structure to a bimodal structure [39].

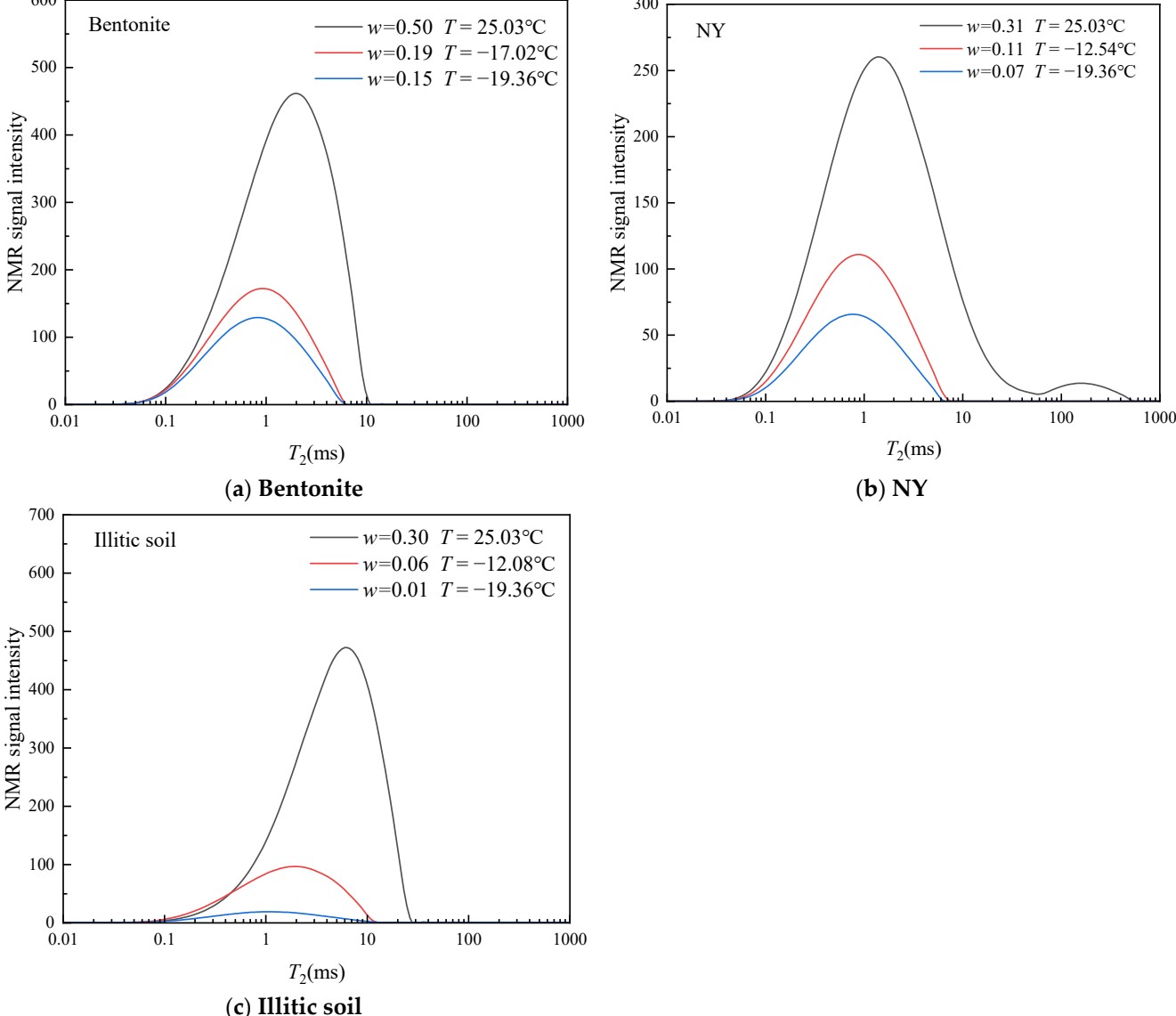

**Figure 4.** $T_2$ distribution curves for bentonite, NY, and illitic soil.

### 3.2. The Influence of Salt Solution Concentration on the Freezing Characteristic Curve

The freezing characteristic curves of the three soil samples measured in the experiment under different NaCl solution concentrations are shown in Figure 5. For soil samples saturated with deionized water, the SFCC during the freezing process can be divided into

three stages: (1) During the freezing process, when the temperature drops from zero to $T_f$, the water content is almost constant, and the soil sample has not yet frozen; (2) Once the temperature drops below $T_f$, SFCC drops sharply, pore water content decreases and a large amount of ice forms, and unfrozen water content decreases. Here, the temperature $T_f$ is the temperature at which the soil sample begins to freeze, defined as the freezing point; (3) When the temperature drops further, the change trend in unfrozen water content slows down and will not decrease significantly. When the temperature drops to $-10\,^\circ$C, a small part of pore water has not yet frozen.

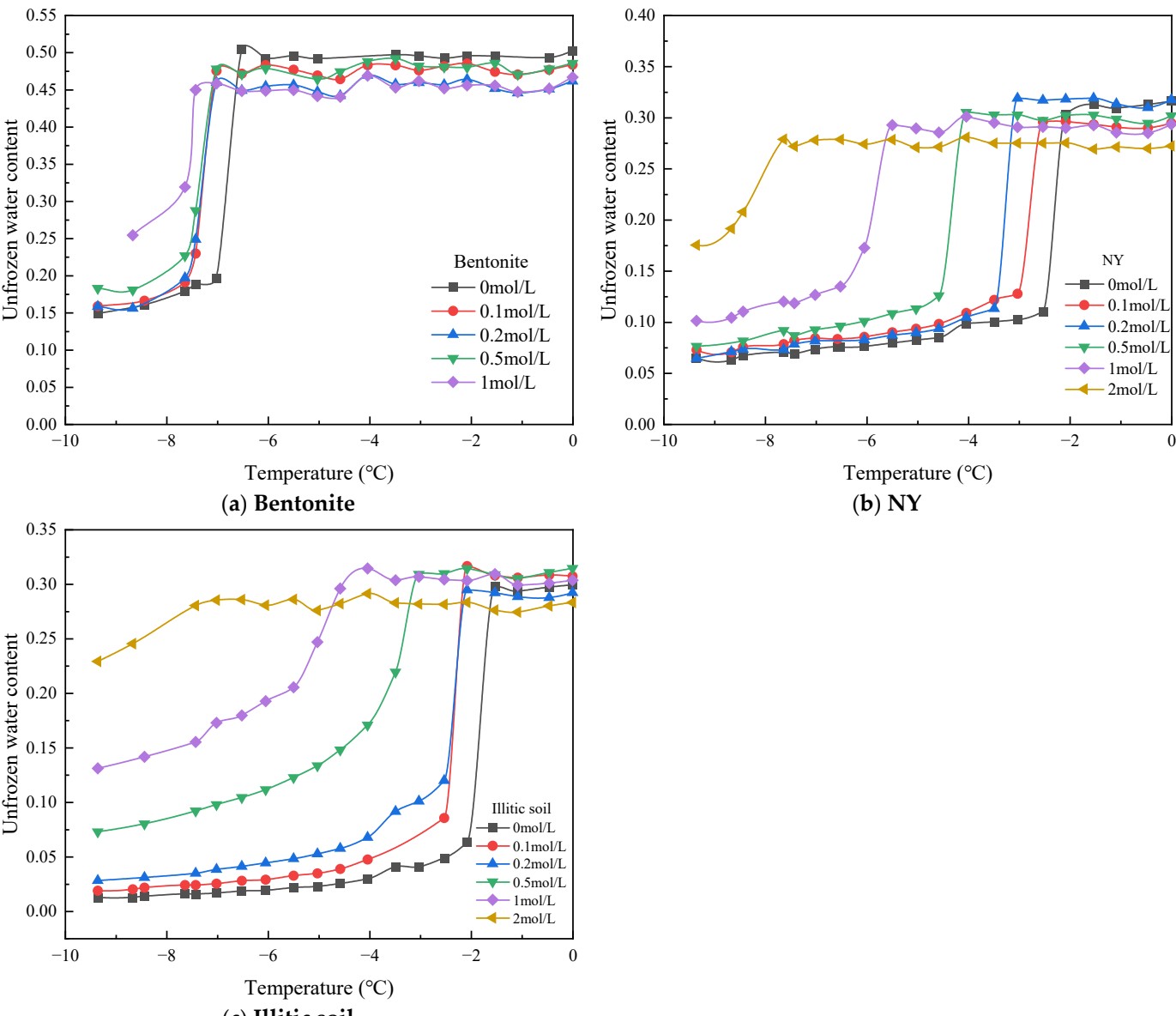

**Figure 5.** Soil freezing characteristic curves for bentonite, NY, and illitic soil under different concentrations of NaCl solution.

The experimental results show that when the concentration of salt solution increases, the freezing point of the soil sample decreases (the curve moves to the left), and the first stage becomes longer. When entering the second stage, when the concentration increases, the speed of curve descent slows down, the slope decreases, and the change in unfrozen water content slows down. That is to say, at the same temperature, the higher the concentration of salt solution, the greater the unfrozen water content in expansive

soils. During the freezing process, as the temperature further decreases, the salt exclusion effect in frozen soil leads to a further increase in the soluble salt concentration in pore water, which has a more significant effect on unfrozen water content. The salt exclusion effect here refers to the exclusion of salt from the pore water when the water freezes into ice, that is, there will be no salt in the pore ice. Therefore, once the soil partially freezes, the salt concentration in the unfrozen water will be higher than the initial concentration. And as the unfrozen water content decreases, the soluble salt concentration increases. The salt solution concentration in the legend is the initial concentration in the saturated soil sample. When the initial salt solution concentration is 2 mol/L, the SFCCs for NY and illitic soils are always in the second stage during the freezing process, that is, unfrozen water content continues to decrease and has not reached the third stage. The bentonite with an initial salt solution concentration of 2 mol/L and three kinds of soil samples with an initial salt solution concentration of 5 mol/L had not reached the freezing point, and the soil samples had not frozen, when the temperature dropped to −10 °C. The study of soil freezing characteristic curves can provide necessary parameters for engineering construction in salinized expansive soil areas in northwestern and northern China.

Figure 6 shows the SFCCs for three kinds of soil samples under condition of deionized water saturation. Among them, the dry density of montmorillonite was 1.2 g/cm$^3$, and the corresponding saturated water content was about 0.5; the dry density values for NY and illitic soils were both 1.5 g/cm$^3$, and the corresponding saturated water content was about 0.3. As can be seen from the figure, as montmorillonite content in the soil sample increased, the freezing temperature of the soil sample gradually decreased, and residual unfrozen water content gradually increased. Following is an analysis of its influence on freezing temperature and residual water content from two perspectives: salt solution concentration and soil properties.

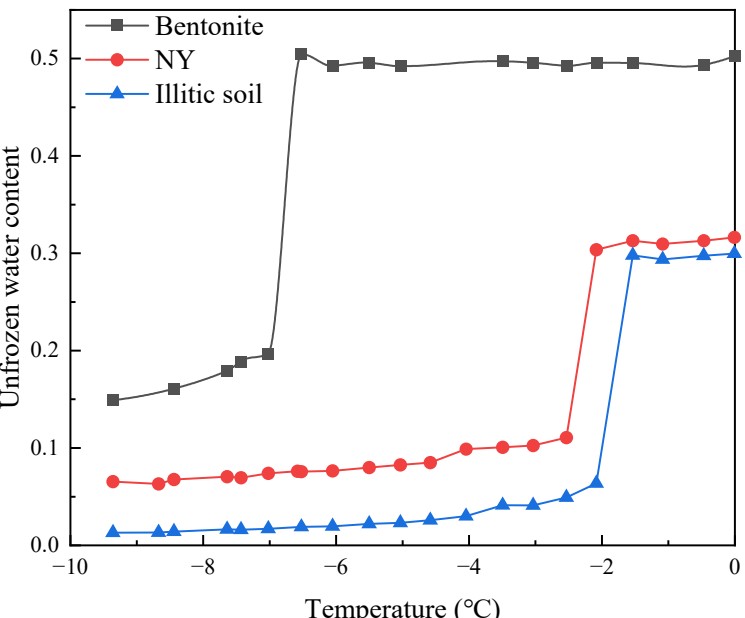

**Figure 6.** SFCCs for bentonite, NY and illitic soil under deionized water saturation.

*3.3. Correlations among Soil Sample Freezing Point, Concentration of Salt Solution, and Soil Properties*

Figure 5 shows the trend in the freezing point of soil samples with the change in salt solution concentration. As the concentration of NaCl increased, the freezing point rapidly decreased. Soil properties also affected the freezing point of the soil samples. When the concentration of salt solution was constant, comparing the freezing points of three types of soil samples revealed that bentonite < NY < illitic soil. The montmorillonite minerals in the soil sample have the characteristic of swelling when exposed to water. Under the premise

that the total volume of the soil sample is constant, the expansion of montmorillonite will compress the large pores in the soil sample into small pores [40,41]. Bentonite has the lowest dry density, so the total pore volume inside the soil sample is the largest, but because it contains a large amount of montmorillonite, its internal pores are compressed into many small voids under expansion. During the freezing process, pore water freezes first in large pores, and then water in small voids will freeze. Therefore, as the content of montmorillonite in soil samples increases, the freezing temperature of the soil samples gradually decreases.

The experimental results show that for the soil samples used in this study, the effect of the salt solution on freezing point depression was more significant than that of capillary action. The freezing point of the soil sample was reduced due to the increase in osmotic pressure in the pore water solution, so the relationship between the freezing point of the soil sample and the salt concentration can be expressed as:

$$T_f + T_{sf} = K_f c_e \tag{1}$$

where $T_f$ and $T_{sf}$ (K) are the freezing points of salt-free soil and salt-containing soil, respectively, $K_f$ (=1.86 K·L/mol) is the freezing point depression coefficient [12], and $c_e$ (mol/L) is the effective ion molar concentration in the salt solution. In an ideal dilute solution, $c_e$ is equal to its actual ion molar concentration c. In a non-ideal dilute solution, $c_e$ is a function of *c*. According to the formula given by Zhou et al. (2018) [12]:

$$c_e = k\ln(1 + \frac{c_0}{k}) \tag{2}$$

where *k* (mol/L) is the effective molar concentration coefficient. When *k* tends to $+\infty$, $c_e$ tends to *c*, and the solution tends to be an ideal solution. Figure 5b shows the effect of the salt solution on the soil freezing point calculated using the above formula, where the black solid line is the calculation result assuming that the solution is an ideal solution (i.e., $c_e = c$), and the black dashed line is the calculation result for the effective concentration using Formula (2). When the concentration is low, the freezing point decreases linearly with the increase in ion molar concentration, which means that the NaCl solution can be assumed as an ideal dilute solution at this time. However, when the concentration of the NaCl solution further increases, the relationship between the freezing point and the ion molar concentration deviates from linearity and shows non-linear changes, which means that the NaCl solution should be assumed to be a non-ideal dilute solution at this time. Therefore, Formula (2) is used to correct the ion molar concentration. Appropriate parameters were used for different soil samples, as shown in Table 3.

**Table 3.** Calculation parameters for bentonite, NY and illitic soil.

| Soil Type | *k* | $T_f$/°C |
|---|---|---|
| Bentonite | 0.7 | −6.5 |
| NY | 0.7 | −2 |
| Illitic soil | 3 | −1.5 |

Substituting Formula (3) into Formula (2) yields the calculation results shown in Figure 7, which are generally consistent with the experimental results.

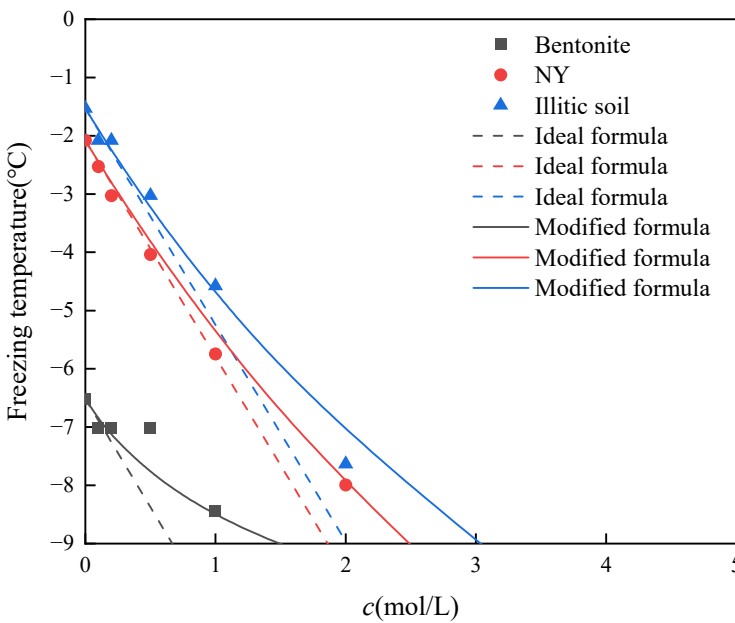

**Figure 7.** Variation trend in freezing temperature of bentonite, NY, and illitic soil with salt solution concentration.

*3.4. Influence of Salt Solution Concentration and Soil Properties on the Residual Unfrozen Water Content*

Figure 8 shows the unfrozen water content in three soil samples under different salt solution concentrations. When the temperature drops to −10 °C, a fraction of the water still has not frozen, which is referred to as the residual unfrozen water content in this experiment. Figure 8 shows the residual unfrozen water content corresponding to three types of soil samples under different salt solution concentrations. As the concentration of the NaCl solution increases, the residual unfrozen water content increases. When the concentration increases to 5 mol/L, the residual unfrozen water content is the initial water content of the soil sample. For bentonite, when the pore salt solution concentration is greater than or equal to 2 mol/L, all water in the soil does not freeze. The higher the concentration of the salt solution in the same volume, the less water it contains, so the bentonite in Figure 8 has less unfrozen water content, corresponding to 5 mol/L, than the unfrozen water content of 2 mol/L. Since the dry density of bentonite is different from that of NY and illitic soil, and the pore size of bentonite is different from that of the other two soils, only NY and illitic soil are compared. It is worth noting that when the solution concentration is less than 0.5 mol/L, the unfrozen water content of NY is greater than that of illitic soil. This is because the montmorillonite in NY expands after absorbing water, compressing large pores into small pores. The expansion in illitic soil is weaker, and the pore size remains almost unchanged. Therefore, there is very little residual unfrozen water content in the illitic soil at −10 °C, while there is still some water that has not been frozen in the small pores of NY. However, as the concentration of the salt solution increases, the expansion of the diffuse double layer is inhibited, and the degree of collapse of NY's large pores decreases. As can be seen from the particle size distribution curve, the illitic soil has a higher proportion of soil particles smaller than 3 μm, so when the concentration of the salt solution is high, the illitic soil has more small pores than NY, and at this time, the illitic soil's residual water content is more than that of NY.

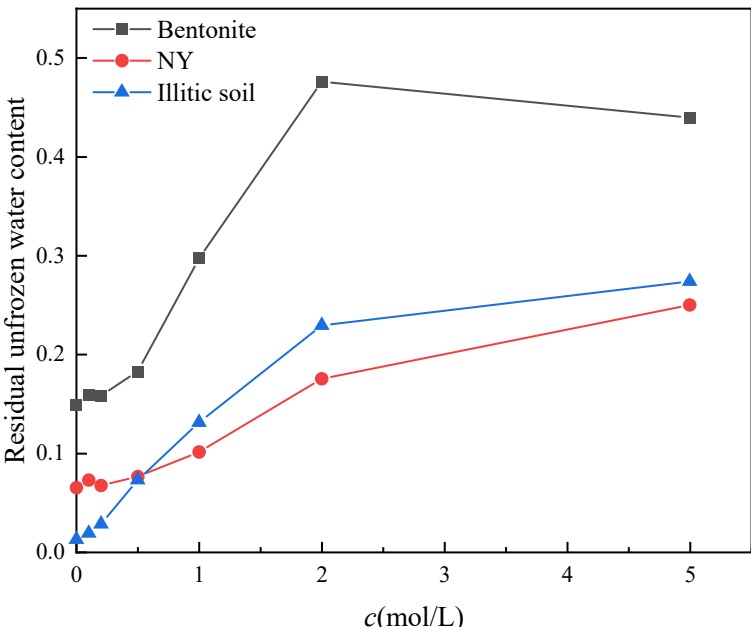

**Figure 8.** Relationship between residual unfrozen water content and salt solution concentration in bentonite, NY, and illitic soil.

## 4. Simulation of Soil Freezing Characteristic Curve

Zhou et al. [12] proposed a generalized Clapeyron equation that can be used to calculate the phase equilibrium state of frozen soil based on the multiphase balance theory model proposed by Wei [42]:

$$\eta(T_0 - T) = p^i(\beta - 1) + S_M + cRT \tag{3}$$

where $\eta \approx 1.23$ MPa/°C; $T_0$ is the freezing temperature of pure water, which is 273.15 K; $T$ is the absolute temperature; $p^i$ is the ice pressure, MPa; $\beta \approx 1.09$; $S_M$ is the matrix suction, MPa; $c$ is the molar concentration of solute, mol/L; R is the ideal gas constant, $R = 8.314$ J·mol/K.

In order to describe the change in unfrozen water content in frozen soil with temperature, the Brooks–Corey model of the soil–water characteristic curve for unsaturated soil can be introduced [12,29,43]. In this experiment, there is no external load on the soil sample, and when it is saturated by deionized water, $p^i = 0$ MPa, $c = 0$ mol/L. Therefore, $S_{M0}/S_M = (T_0 - Ts)/(T_0 - T)$.

$$\frac{w - w_r}{w_0 - w_r} = \left(\frac{S_{M0}}{S_M}\right)^B = \left(\frac{T_0 - T_s}{T_0 - T}\right)^B ; T < T_s \tag{4}$$

where $T_s$ is the freezing temperature of the soil sample; $w_0$ is the initial water content; $w_r$ is the residual unfrozen water content; $S_{M0}$ is the air entry value;

Equation (5) can be simplified as:

$$w = A(T_0 - T)^{-B} \tag{5}$$

In the above equation, $A = (w_0 - w_r)(T_0 - T_s)^B$; $B$ is a fitting parameter, which is related to soil properties.

The fitting of the freezing characteristic curves for the three experimental soils saturated with deionized water using Equation (5) is shown in the Figure 9, and the fitting parameters are shown in Table 3.

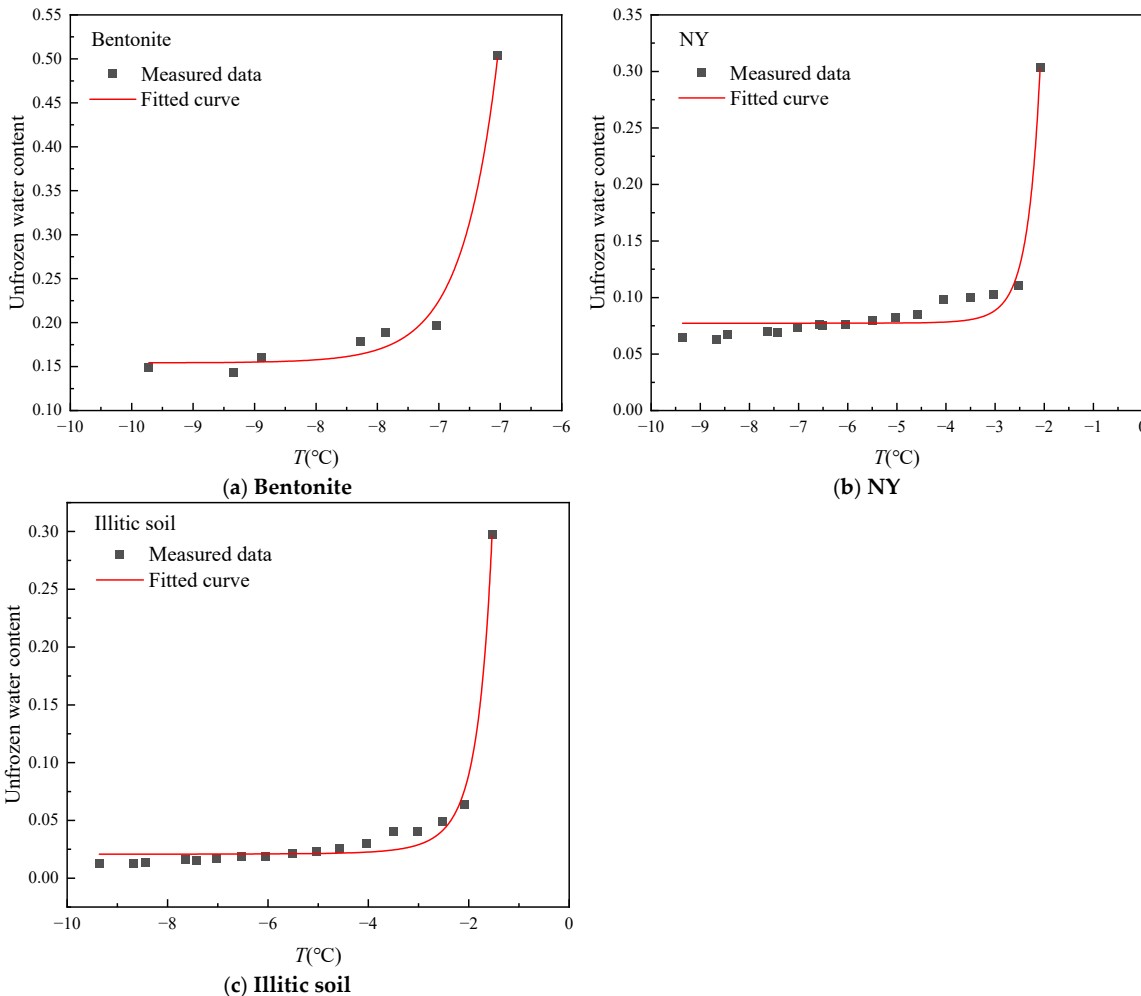

**Figure 9.** Model simulation results for bentonite, NY, and illitic soil.

As can be seen from Table 4, the value of parameter *A* increases significantly with increases in the soil expansibility, indicating that the difference between the freezing temperature of the soil and that of pure water is greater. The Gibbs–Thomson effect [44] points out that when pore water in the soil begins to freeze, the surface tension generated at the ice–water interface in the pores causes the pore water pressure to be less than the pore ice pressure. The capillary water pressure at this time is less than the capillary water pressure before freezing, which leads to a decrease in the chemical potential of capillary water. The higher the content of montmorillonite minerals, the smaller the pores of the soil. The more significant the Gibbs–Thomson effect is, it will lower the freezing temperature of the soil. The value of parameter *B* is related to saturated water content. Bentonite has a lower dry density, corresponding to a saturated water content of 0.5; both NY and illitic soil have a saturated water content of about 0.3. Therefore, bentonite has the largest B value, while NY and Illitic soil have similar B values. In addition, the higher the montmorillonite content, the larger the residual unfrozen water content $w_r$ is. This is because there are many micro pores in montmorillonite minerals.

**Table 4.** Model parameters of Bentonite, NY, and illitic soil.

| Soil Type | *A* | *B* | $w_r$ | $R^2$ |
|---|---|---|---|---|
| Bentonite | $6.94 \times 10^{20}$ | 26.16 | 0.148 | 0.98 |
| NY | 90.36 | 8.19 | 0.077 | 0.96 |
| Illitic soil | 2.55 | 5.23 | 0.02 | 0.98 |

Zhou et al. [12] derived a differential relationship between water content and temperature in salt-containing soil without external load based on the generalized Clapeyron equation:

$$\frac{\mathrm{d}w}{\mathrm{d}t} = \frac{n\rho Rc_0 w_0 w + \eta w^2}{f(w)w^2 + n\rho RTc_0 w_0} \tag{6}$$

$$f(w) = \frac{\eta}{AB}\left(\frac{w - w_r}{A}\right)^{-\frac{B+1}{B}} \tag{7}$$

In the equation, $n$ is the number of ions produced by a solute molecule in the solution (for NaCl solution, $n = 2$), $c_0$ is the salt concentration of the pore solution before soil sample freezing, and $\rho$ is the density of pure water.

Therefore, under the premise of known initial water content, using the BC model to obtain the parameters of the soil sample freezing curve under pure water conditions, we can obtain parameters $A$, $B$, and $w_r$. This allows us to calculate the freezing characteristic curves of soil samples saturated with different salt solution concentrations. Based on this idea, taking a temperature step $\Delta T = -0.1$ K, we used the improved Euler method to calculate for the three soil samples tested in this paper and compared with the experimental results. The results are shown in Figure 10.

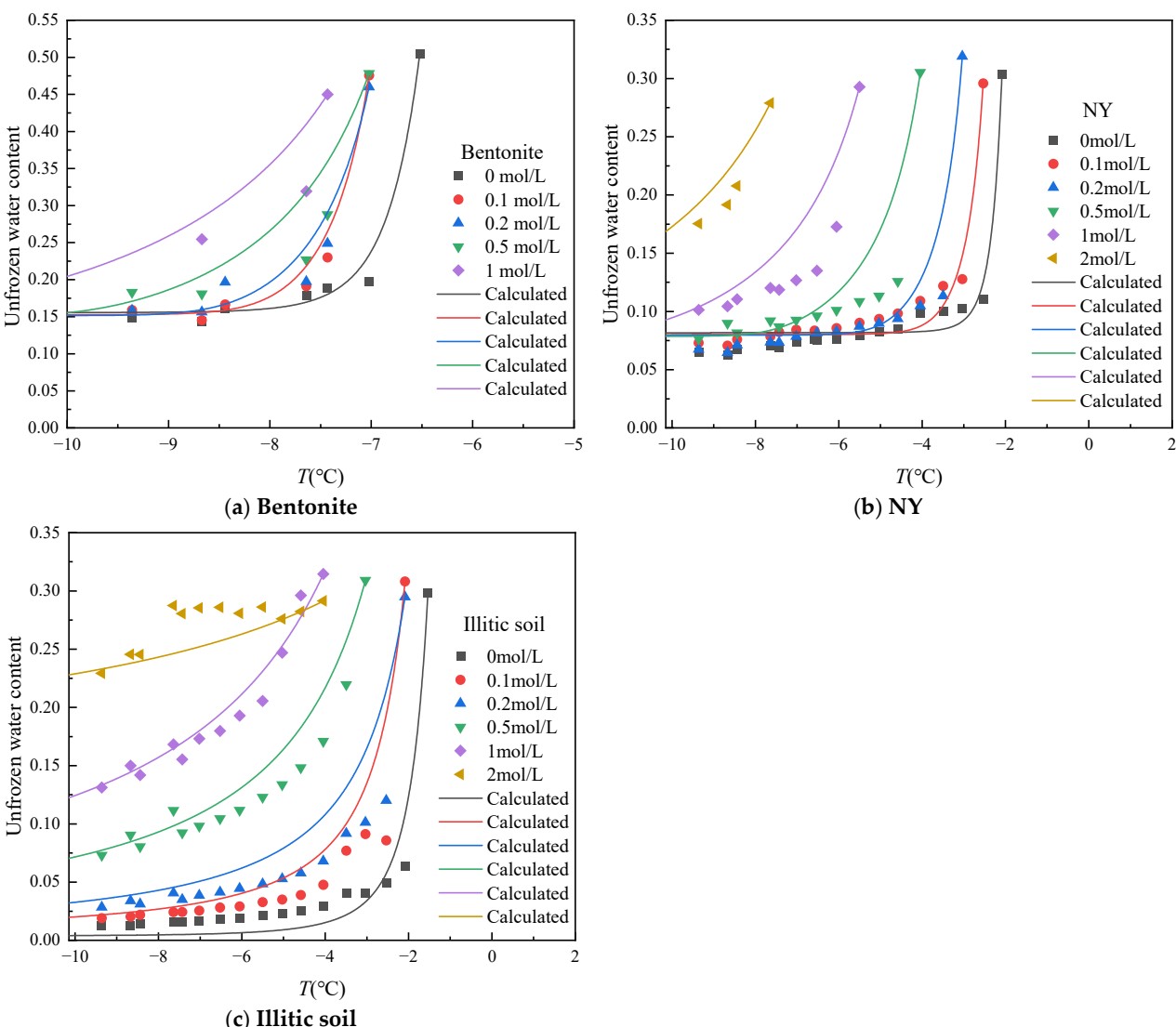

**Figure 10.** Comparison between model calculation values and experimental results.

Table 5 shows the coefficient of determination of the model-calculated values to the experimental data. The results indicate that the calculation results of Zhou's model are in overall agreement with the experimental results in this paper, demonstrating the applicability of this model in predicting the freezing characteristic curve for unloaded saline soil.

**Table 5.** Coefficient of determination between the calculated values of the model and the experimental data.

| Solution Concentration /(mol·L$^{-1}$) | Coefficient of Determination | | |
|---|---|---|---|
| | **Bentonite** | **NY** | **Illitic Soil** |
| 0 | 0.98 | 0.96 | 0.98 |
| 0.1 | 0.98 | 0.96 | 0.89 |
| 0.2 | 0.94 | 0.91 | 0.89 |
| 0.5 | 0.87 | 0.83 | 0.95 |
| 1 | 0.79 | 0.86 | 0.96 |
| 2 | - | 0.96 | 0.87 |

## 5. Conclusions

This paper presents experimental research conducted on the effects of salt solutions and soil properties on the freezing characteristic curves for expansive soils. A cold bath was used to control the freezing temperature of soil samples, and the unfrozen water content in the expansive soils was measured by nuclear magnetic resonance technology. The following conclusions were drawn:

(1) The freezing characteristic curve for expansive soil is very similar to the water retention curve. When the temperature reaches the freezing point, the soil sample begins to freeze; as the temperature further decreases, a large amount of pore water begins to freeze, and the unfrozen water content rapidly decreases; when the temperature further decreases, the speed at which the unfrozen water content decreases slows down.

(2) When soil samples are saturated with salt solutions of different concentrations, the freezing point of the soil decreases with increasing salt concentration. At a given negative temperature, there is more unfrozen pore water in saline soil. Therefore, under the same conditions, the presence of salt promotes thawing in frozen areas. During thawing, saltwater freeze zones provide more liquid water than non-saltwater freeze zones. The higher the montmorillonite content in soil samples, the lower the freezing temperature of the soil samples and the higher the residual unfrozen water content. This conclusion can explain the existence of winter thawing areas in Nanyang.

(3) Based on the similarity between soil freezing characteristic curves and soil–water characteristic curves, a unified model in the literature was used to simulate the experimental results for the three soils under different salt contents. Simulation results show that freezing characteristic curves under different salt contents can be simulated using one set of parameters, indicating that the effects of the salt solution and capillary action are independent. At the same time, when the salt solution concentration is large, it deviates from the ideal solution assumption and requires correction.

**Author Contributions:** Conceptualization, T.M. and H.Y.; methodology, P.Y.; software, F.H. and Q.Z.; validation, F.H., H.Y. and T.M.; formal analysis, H.Y.; investigation, T.M.; resources, P.Y.; data curation, H.Y.; writing—original draft preparation, H.Y. and Q.Z.; writing—review and editing, T.M.; visualization, H.Y.; supervision, T.M.; project administration, T.M.; funding acquisition, T.M. and P.Y. All authors have read and agreed to the published version of the manuscript.

**Funding:** This research was funded by National Natural Science Foundation of China (41972290), the Science and Technology Department of Guangxi (No. AD20325010) and National Natural Science Foundation of China (12002243).

**Institutional Review Board Statement:** Not applicable.

**Informed Consent Statement:** Not applicable.

**Data Availability Statement:** The research data can be provided upon request to the corresponding author.

**Conflicts of Interest:** The authors declare no conflicts of interest.

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
