# Peer review of "Experimental Investigation on the Effect of Salt Solution on the Soil Freezing Characteristic Curve for Expansive Soils"

_sustainability, doi:10.3390/su16010363_

Round 1
Reviewer 1 Report
Comments and Suggestions for Authors
Review
Experimental investigation on the effect of salt solution on the soil freezing
characteristic curve of expansive soils
Haiwen Yu, Fengfu Hao, Panpan Yi, Qin Zhang and Tiantian Ma
General:
The manuscript can be shortened a little because it has repetitive parts or known from other papers. It needs major changes.
It requires more intensive bibliographic control and complementation of data (granulometries, minerals, salt concentrations/% in brines trials).
I think that at some points the authors ignore many previous laboratory experiences that involve the basic concepts of this work. They should make a short history of the scope of discoveries like Tsytovich and Anderson/Morgenstern that have been based on many other
authors and summarize in the direction they want to know. There are not enough citations in this regard; the authors are kindly invited to review this concept.
A discussion is needed between the tests with these materials and those that came from saline soils.
I propose removing some paragraphs because they contain truisms (I have marked some as a suggestion)
Abstract
Specify briefly which soil properties are being referred to, this topic is repeated without mentioning more specifically which ones are being taken under which experiments. It should be mentioned
L 22-23 “the concentration of salt solution increases, the freezing temperature of the soil decreases; The topic is known, improve the sentences, with quotes later,
Introduction
1. L 34, I would replace seasonal permafrost by seasonal frost (e.g. Washburn,
1979, Trombotto Liaudat et at., 2014),
2. L 38 “High temperatures”, must be replaced by above 0 °C, mention those
temperatures that you saw by the soil subsidence, add a fig with a curve
3. L 62-66 The topic is very well known, I would cite other Western specialists as well, see Anderson and Morgenstern, etc.
4. L 64-65. Improve…Solutes lower the freezing point of water. The freezing-point depression in dilute solutions is given by the Van't Hoff equation (Lewis and Randall 1961, see Marion, 1995, etc.),
I do not believe that the influence of salts such as sodium chloride in periglacial environments is little known. I believe that authors should consult more Western literature, as such as the experiences at the CRELL in the USA.
I think they should put emphasis on their own studies and conclusions from the Nanyang area, which is new and I unfortunately do not know.
Materials and Methods
Explain better the mineral composition of the materials, for example, sodium bentonite, which seems to have no Ca? Or that commercial illitic soil, that is not well known for everybody. The Nanyang soil has a very high clay content but not only montmorillonite, correct? What is that?
Results and Discussion
There is a lack of discussion mentioning important authors who worked on the topic. Compare, for example, with other examples of bentonites, as in the freezing and thawing cycles can originate other phenomena that influence the SFCC, such as the aggregates that form.
Point out the soil freezing characteristic curve of expansive soils in Nanyang, and what are the news and contributions for other areas on Earth.
Figures
The figures are explanatory although the titles do not always express the content or all information. I suggest using the same units in the figures
References
Important authors on the subject still need to be mentioned
Please see more comments in the manuscrip

Reviewer 2 Report
Comments and Suggestions for Authors
This manuscript investigates the effects of salt solution on the SFCC, and a model is proposed to predicate it. The research is interesting to understand the change characteristics of unfrozen water of frozen soil. The manuscript is well organized, and the current from can be improved according to the following comments.
(1) Line107-121: need to show why to choose the water content as 20% and the dry density of 1.5g/cm3 and 1.2g/cm3.
(2) Figure 5: the vertical axial name will be better using 'unfrozen water content'
(3) Figure 5: please show the reasons that why the water content will not change when the temperature is greater than -6℃ for the 0mol/L as shown in figure 5a.
(4)Figure 10: This reviewer encourages to show the error analysis for the test data the calculated data.
(5) Line 178: should be Figure 4.
(6) Line 29: should an English comma.
Comments on the Quality of English LanguageNeed be improved.
Reviewer 3 Report
Comments and Suggestions for Authors
1. The findings described in the abstract and conclusions of the manuscript are basically common sense perceptions, and many of the results are consistent with our conventional perceptions. It is recommended to further condense the innovative results and even quantitative conclusions in this study. Therefore, the abstract as well as the conclusions should be rewritten.
2. It is not clearly explained why the 3 types of experimental materials including medium expansive soil (NY) from the Nanyang area, sodium bentonite, and commercial illitic soil are selected in this research.
3. Figure 8 is not cited in the text.
4. Please explain why the curve of residual unfrozen water content and salt solution concentration for bentonite decreases when the c is higher than 2 mol/L in Figure 8.
5. The units of the variables including SM and c should be given in equation (4).
6. The English of the whole manuscript should be improved.
7. It is still hard to understand why the dry density of the bentonite is different from NY and Illitic soil.
8. The research background is not clearly stated. For example, why the effect of salt solution on the soil freezing characteristic curve should be studied?
Comments on the Quality of English LanguageThe English writting is better to be improved.
Round 2
Reviewer 1 Report
Comments and Suggestions for Authors
The paper has been efficiently improved.
The bibliography can still be enriched but I understand that not all of it is accessible everywhere. I suggest including other observations in the experiments, for example, see:
Batbaatar, Jigjidsurengiin; Gillespie, Alan; Sletten Ronald; Mushkin, Amit; Amit, Rivka; Trombotto Liaudat, Dario; Liu, Lu and Petrie, Gregg. 2020. Toward the detection of permafrost using land surface temperature mapping. Remote Sens., 12(4), 695; https://doi.org/10.3390/rs12040695 - 20 Feb 2020
In the future, experimental observations should be carried out in areas with permafrost.
Author Response
Thank you very much for your advice.
We have added information on the application of remote sensing technology in observing permafrost regions, and enriched our bibliography. We have cited the following article.(See Lines 40-44)
Batbaatar, Jigjidsurengiin; Gillespie, Alan; Sletten Ronald; Mushkin, Amit; Amit, Rivka; Trombotto Liaudat, Dario; Liu, Lu and Petrie, Gregg. 2020. Toward the detection of permafrost using land surface temperature mapping. Remote Sens., 12(4), 695